# The Human–Animal Relationship as the Focus of Animal-Assisted Interventions: A One Health Approach

**DOI:** 10.3390/ijerph16193660

**Published:** 2019-09-29

**Authors:** Lucia Francesca Menna, Antonio Santaniello, Margherita Todisco, Alessia Amato, Luca Borrelli, Cristiano Scandurra, Alessandro Fioretti

**Affiliations:** 1Departments of Veterinary Medicine and Animal Production, Federico II University of Naples, 80138 Naples, Italy; antonio.santaniello2@unina.it (A.S.); margotodisco@yahoo.it (M.T.); alessiaamatovet@gmail.com (A.A.); luca.borrelli@unina.it (L.B.); alessandro.fioretti@unina.it (A.F.); 2Department of Neurosciences and Reproductive and Odontostomatological Sciences, University of Naples Federico II, 80138 Napoli, Italy; cristiano.scandurra@unina.it

**Keywords:** animal assisted intervention, one health, interspecific relationship, interspecific attachment, zoonosis risks, safety of care administration

## Abstract

Background: Animal-assisted intervention (AAIs) represent an adequate expression of integrated medicine, according to the One Health approach. We argue that AAIs are interventions based on interspecific relationships between humans and animals. Although there are many studies on the effects of AAIs on animal and human health and wellbeing, research is still needed to give us more data. For example, information is still lacking on the aspects characterizing and influencing the interspecific relationships occurring in AAIs. The efficacy of an intervention based on interspecific relationships will be influenced by different factors, such as attachment styles and personalities of both the animal and the handler, an appropriate choice of animal species and their individuality, animal educational training techniques, the relationship between the handler and the animal, and relational reciprocity between animal, the patients, and members of the working team. Method: This article aims to contribute to the study of interspecific relationships in AAIs via theoretical considerations. An interspecific relationship determines the result of safe interventions, which directly influences the welfare of the animal. Results and considerations: AAIs should be evaluated systemically as a network within a process in which every component interacts with and influences other components. Standardized methods using appropriate tests and parameters are needed to better select appropriate animals (i.e., species and individual subjects) using interspecific relational competences as well as appropriate educational training methods and health protocols to assess potential risks.

## 1. Background

The limitations of some pharmacological therapies safeguarding against specific diseases, such as Alzheimer’s, autism, and Parkinson’s, have led researchers to evaluate alternatives, with a particular interest being on animal-assisted therapies. These are considered to be co-intervention therapies supporting conventional therapies [1,2,3,4]. Integrated medicine, however, concerns both the prevention and treatment of diseases via a biopsychosocial approach [5,6,7,8]. Using this approach, we can consider a healthcare zooanthropology vision [9], which studies and applies an interspecific relationship between humans and animals in both healthcare/therapeutic and educational contexts. This view supports International Association of Human-Animal Interaction Organizations (IAHAIO) definitions for animal-assisted intervention (AAI) and guidelines [10] and represents an effective expression of health according to the One Health *principium* [11,12,13].

The close behavioral correspondence between humans and animals is widely recognized [14], and humans, such as other mammals, develop the necessary attachment bonds which create a relationship [15,16,17]. In addition, the *Biophilia* concept, which is defined as the instinctive and necessary attraction of humans towards the animal kingdom, is one of the complex aspects to consider in understanding the role of animals on human psychophysical well-being [18,19,20].

Stimulated by the observations of Hediger et al. [21], this contribution proposes our point of view and some suggestions on methodological approaches and the application of AAIs.

## 2. AAI Interspecific Relationships and Attachment Bonds

The effectiveness of AAIs and therapies will depend on the relationship between the handler, the animal, the patients, and any other healthcare professionals involved in the AAI. Humans, like other mammals, develop a necessary bond of attachment that will condition their way of relating. As relationships and, above all, interspecific relationships represent the focal point of AAIs, attachment styles play a primary role when therapeutic goals need to be achieved. This is different when the goals are more educational (AAE—animal-assisted education) or supportive (AAAs—animal-assisted activities), in which case it is not really plausible to expect a change in this area. It is, however, very plausible to assume that even in AAE and AAAs, the attachment bond plays a crucial role as all significant relationships activate respective internal working models. Thus, we can assume that this also occurs within the dyad, or working pair, of the animal handler and the animal during educational training. This attachment perspective is the framework which supports all kinds of relationships, including interspecific ones that have been studied in a comparative perspective way [22,23,24,25,26] and has already been fruitfully applied to AAIs. To this end, Payne et al. [26] reported that the human–dog relationship has much in common with a parent–child relationship and satisfies the four prerequisites for an attachment bond (proximity seeking, secure base, safe haven, and separation distress).

Notwithstanding these premises, recent studies are questioning the complete validity of the equation between dog–human attachment and infant–parent attachment [27,28], asserting that the dog–human attachment is likely to be closer in reality to adult pair bonds or friendships, in which attachment represents only one component of a multifaceted relationship. In addition to attachment bonds, it is fundamental to consider other potential mediating or moderating variables that are able to produce psychological or physiological changes (e.g., the reduction of stress, anxiety, and aggression or the promotion of trust, motivation, and concentration) thanks to AAIs, such as social support or distraction processes [29].

The AAI represents a very complex system of relational feedbacks that start with bodily gestures and attitudes and with the activation of emotional sense–motor models between the two species. Based on scientific literature [30,31,32], different animal species are involved in AAIs but more than other species, dogs seem to facilitate the construction of a relationship in which reciprocity is greater, and the therapeutic work is more profound [32]. Moreover, thanks to its ethological characteristics, the dog not only learns through games, just like children but also allows for the establishment of an active relationship, communication, and interaction [33]. The intervention, in fact, of the all different types of AAIs, consists in the structuring of games that the handler, with the advice of an experienced doctor or psychologist, performs based on the species involved, the individual characteristics of the animal, and the purpose of the intervention.

## 3. Suggestions for Research Actions

### 3.1. The AAIs as “Complex Systems”

According to Hediger et al. [21], which reports the need for a systemic approach in studies of the interrelation and reciprocal influence of the relationship between participating humans and animals and included environmental factors and social contexts. We strongly consider and encourage the study of an interspecific relationship as reciprocity. A relationship is a complex system [34,35,36,37,38] in which a circularity represents the essential characteristics, whereby all components influence each other. According to which, the approach to a complex system cannot be understood if one proceeds with an analytical view through a simple breakdown, trying to examine every single part. In this way, complex systems are differentiated from complicated ones. Complex systems are represented above all by living organisms and continually respond to the law of dynamics and change due to internal pressures within the same system. Complicated systems, on the other hand, are represented by electronic instruments that are immutable in that they are always equal to themselves. They are orderly and predictable because they respond to the laws of cause and effect (the computer, for example, is the quintessential complicated system). A complex system can only be explained if you try to obtain a vision of the whole while verifying how much the single parts influence each other reciprocally.

This approach has influenced many areas of knowledge, from neuroscience to information science, linguistics, and psychology. The AAIs are a system within which there are relational dynamics of living beings belonging to two different species. AAIs are a complex system with all the parts interacting and influencing each other to make a result that is greater than its parts. The result is an improvement in the characteristics of the patient. Most medical assessments work with linear systems which lend themselves to assessment in exact, mathematical ways. Therefore, new ways of assessing AAIs are needed. The strength of these interventions lies in the interspecific interaction that involves the emotional, cognitive, linguistic, and behavioral dynamics of the parties involved, among which there is an animal, often a dog, which enriches the complexity because it brings its own languages and dynamics. The approach to the study of relationships, above all, when aimed at therapeutic activities (i.e., animal assisted therapies, AATs) and even more so in relation to interspecific interactions, requires an observation that recognizes complexity and only, in this case, would it be possible to fully understand the dynamics and the effects. A therapeutic relation, such as that of an AAT, is found to be a dynamic system influenced by many variables which condition its evolution. The emergent behavior of a system is due to its nonlinearity. The properties of a linear system are additive—the effect of a group of elements is the sum of the effects considered separately and then as a whole. New properties do not appear which were not already present in the single elements. But if there are combined terms/elements which depend on each other, then the complex is different from the sum of the parts, and new effects appear [39]. Therefore, because of its characteristic intrinsic modality, an AAI fully responds to all of the requisites referring to complex systems, including self-organization. Self-organization is the tendency of a system to generate new forms and structures and starts with an internal dynamic, chaotic behavior, which is the tendency of a system to oscillate between moments of order and moments of disorder without compromising the cohesion of the system. This cohesion is characteristic of the dynamic that takes place within all relationships (AAIs) and, above all, in therapeutic relationships (AATs). On the contrary, disorder rather than order is the generator of new evolutionary possibilities. Unpredictability is strictly linked to chaotic behavior. The action of animals does not follow predictable laws of dosage–response, as the effects of drugs that are proportional to their dose, but it varies in a space–time sense because it can have different effects on different individuals and on the same individual at different times. The action of the animal expresses emergent properties, which are realized in complex systems, as opposed to simple or complicated ones. In this case, new and unexpected properties emerge that are not deducible from the sum of the elementary components.

In AAIs and above all in AAT, they represent the surprising and often unexpected results observed. Circular causality is another characteristic of complex systems and is the opposite of linear causality. The causes and effects are confused, and phenomena of action and retroaction are created, resulting in the creation of vicious and virtuous circles. AAI is the expression of feedback and circular causality of the complex systems which represent it, namely the patient, the professional team, and the animal. It is precisely that emotional resonance, the variability of the response as well as the unquantifiable and unmeasurable action, which makes AAI an expression of a complex system and it must be considered and studied as such. Within an AAI setting, this interspecific relationship and its circularity influence the results of the intervention, its safety, and the welfare of all components (the animal, the operator, and the patient). When we consider an AAI setting, the complexity concerns the interrelationship of different variables that, both at an individual and combined level, play a fundamental role in the outcome achievement (e.g., a change in the patient’s behavior, experiences or characteristics occurring after the intervention) through the action of the dynamic process (e.g., a therapeutic or educational bond, self-relatedness, temporal patterns, etc.). Within an AAI setting, we should consider at least four parts of the system—the animal, the operator, the patient, and the setting itself.

Considering the wide variety of studies and reports on the effects of AAIs in a dose–response and cause–effect binary logic manner, we suggest that these interventions should be observed as a dynamic network where the relationships between the elements are the main parameters to be evaluated. This means that future studies should assess not only AAI outcomes but also the processes which allow them to reach specific outcomes. This can be both through quantitative and qualitative procedures as well as through personal and clinical reports to collect and assemble data received from different methodologies (quali-quantitative) and perspectives (e.g., therapist vs. patient) (Appendix A).

### 3.2. Choice of the Animal

Using this logic of circularity and relational reciprocity, we can choose both the species and the individual animal for the AAI. In our opinion, the choice of the species depends on their relational competence, which derives from the history of coevolution with human beings, the ethogram, and the breed [24,40,41,42,43,44,45,46,47]. During the domestication process, animals have come to develop a deep relationship with humans. Their evolutionary history with man influences the reciprocity of the relationship and their ethological competences in relation to man. Since prehistoric times, humans have only succeeded in domesticating approximately 20 different animal species. Archaeological evidence suggests that dogs were the first species to be domesticated [45,46,47]. Men and wolves established a relationship of a mutual alliance between predators [48], while in other species (e.g., Equidae) such a relationship was created via man’s predatory actions [40,41]. The search for meat may have been an initial motivation in the very early stages of domestication, meaning that horses progressively became important tools for transportation.

There are also relational differences between each species, for example, Equidae are able to determine a relationship based above all on physical contact. On an ethological level, Equidae are looking for a simple proximately closeness while humans seek real physical contact. Therefore, reciprocity must be created via appropriate manipulations and training as, being a species of prey, reciprocity is not spontaneous for horses on an ethological basis. From the literature consulted, it emerges that the method of approach and manipulation are more important for the horse than the person as horses have the same responses even when an unknown person handles them [40].

On the other hand, the coevolution of the cat with human beings is complex. The cat has evolved as a carnivore with a solitary lifestyle, and in many contexts, it does not need to develop social signals and may not even like the presence of man. Cats do not have a large repertoire of visual communications, and it is quite probable that a cat would become inactive, hide and inhibit habitual behavior [41] within the restrictive conditions, transportations, and new environments that characterize AAIs. Literature on the effect of interactions with cats is still very scarce.

In our opinion, it would not be advisable to work with unconventional pets, such as rabbits and ferrets, among others, as they are not recognized as highly relational animals on an ethological level. With these animals, pleasure in a relationship is usually expressed exclusively towards their owner, and they are unlikely to open up to polyvalent relationships. This would make them vulnerable and passive in an AAI setting as lack of body communication, or non-participation would be an undesirable element and even contradictory to the therapeutic goals set [22]. Furthermore, parameters of stress/well-being are not yet standardized as well as the health issues relating to zoonotic agents, of which they can be carriers, could expose to situations of risk the veterinarian, having legal responsibility for safety and taking charge of the animal, the handlers (AAA and AAE), and the people involved in the AAIs.

Dogs, on the other hand, are one of the main species involved and studied in AAIs [30,31,49] as they have an important competence—the ability to read the non-verbal language of the human being [50,51,52]. Dogs have developed a particular preference for humans, and the ability to recognize conspecifics seems to have played an important role in their genetic changes [53]. From the literature consulted [54,55,56,57,58,59,60], it emerges that the dog understands the difference between the owner and other people. Studies by Prato-Previde [57,58] and other authors [59,60] state that the dog shows greater confidence in new stimuli in the presence of its owner. Observing the link between dogs and owners, it has been shown that relational factors influence the performance of the dog [61,62]. This type of relational dimension offers the dog an important sense of security, and it is able to express itself and neutralize the stress of the first meeting until the next stage of adaptation [24]. Horses can be subject to work stress deriving from contradictory commands or demands from the handler for strong control over their emotions [40]. In our working model, the animal species of choice is the dog, and according to our operative model, the choice of dog should be made by testing its personality with the Monash test that is useful to highlight the dog’s relational abilities combined with the Neo PI-III test on the handler’s personality [63,64]. Some authors show this reliability [65,66,67]. We gave importance to interspecific relationships between the dog and the veterinarian/handler/owner to obtain a balanced and harmonious relationship for both species. Moreover, it is the veterinarian/dog dyad that is paired with the patient and not just the animal in the AAT and in the AAA for the healthcare facilities, instead of the handler/dog dyad in other settings. The expert AAIs’ psychologist chooses the dyad based on the relational characteristics most useful for each AAI setting [22,24]. We emphasize the need for more in-depth studies on the relationship competences of other species during AAIs (in particular the Equidae) and to prepare tests that are able to accurately evaluate them. These studies are necessary to form suitable dyads with defined parameters in settings that involve other species too.

It is necessary to bear in mind that every animal species has different ethological characteristics and relational competences, which depend on their coevolutionary process. If these characteristics and competences are taken into consideration when choosing an animal species, we are able to carry out an AAI to the best of our ability. In this way, the welfare of the animal is protected from a bioethical point of view in which the animal is actively involved but in no way exploited.

### 3.3. Zoonotic Risks in the “One Health” Approach: Safety of Care Administration (Decrees on Safety at Work Given the Legislative Decree 15 August 1991, n. 277, Concerning the Implementation of Directives n. 80/1107/CEE, n. 82/605/CEE, n. 83/477/CEE, n. 86/188 / EEC, and n. 88/642/EEC) [68]

Since the early 2000s, the concept of zoonoses has changed and now includes not only infectious agents but also any harm to humans resulting from an interaction with an animal and vice versa [69,70]. It is possible that patients/users, just like owners and their animals, could be exposed to a zoonotic risk during interventions [71]. The risk factors relating to the animal include species, age, breed, relationship skills, relationship with the handler, style and personality of the attachment, health and behavioral status, previous medical and behavioral history, educational training program, and the ability of the animal handler to remove or intervene promptly to minimize the effects of stress [72,73]. The risk factors relating to the patient/user include age, type and level of pathology, correct functioning of the immune system, attachment style, and any aversion or individual allergy. Other risk factors include environmental ones, such as structural materials (washable, disinfect-able, slippery, etc.), the level of information and involvement of personnel, temperature, odors, and levels of hygiene/health. For these reasons, a predictive risk analysis that evaluates the probability of a negative event manifesting itself in a specific context is needed. This approach aims to define a standardized health protocol for different types of interventions that involve different species of animals according to the ethological abilities of the animal [24]. The possibility that an animal could be at risk due to human dynamics is always considered. Prevention starts with an initial selection based on the behaviorist veterinarian’s visit and adequate training. Standardized educational training programs should be followed to allow the animal to express its abilities and individual skills, thus creating a positive and enjoyable experience during interventions and reducing any zoonotic risks. An interspecific harmonic relationship is recommended, in which the human referent represents a safe base for the animal.

Furthermore, zoonotic risks must also be assessed in light of European directives on patient safety and the quality of medical care to comply with current legislation [68] and according to international recommendations of The Society for Healthcare Epidemiology of America (SHEA) [74]. SHEA is a professional society that improves public health by establishing infection-prevention measures and supporting antibiotic stewardship among healthcare providers. [74]. Banach et al. [75] have prepared useful recommendations for the prevention and management of risks in health facilities. Murthy et al. and Hardin et al. [76,77], in particular, focused on reducing the potential risks associated with the presence of animals in health facilities; we want to add our contribution by focusing attention on legal security. In the AATs and all AAA in the healthcare facilities, where both the animal and patients and workers are most at risk, it would be preferable if the handler of the animal was an expert veterinarian who is able to take care of the animal’s health and assess any zoonotic risks in real-time. This would not only protect the human beings and participating animals during the intervention from a legal point of view but also the health facilities that host them. [4,22,24]. In 2014 [68] European Member States were invited to encourage professional healthcare organizations to create a collaborative, inter-professional health culture for patient safety [78]. AAIs, with their interdisciplinary character, fully interpret this need by demonstrating that they represent the new frontiers of integrated medicine and research [79].

Through these reflections, we hope to contribute to a new, inter-professional, cultural change.

## 4. Conclusions

Scientific literature shows that AAIs are increasingly attracting interest but still require further study to address the specific, standardized aspects that we consider essential. Parameters that need to be further investigated include interspecific relationships and the factors influencing them, as well as the inter-specific relational competences of the species and individual animal chosen and its relationship with the handler. All of these factors will influence the welfare of the animal, the final outcome of the intervention, and their safety. We look forward to the standardization of animal educational training programs and specific tests to better choose a suitable animal for AAIs and to carry out a risk analysis to define standardized specific healthcare protocols for different kinds of interventions.

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
