# Peer review of "The Human–Animal Relationship as the Focus of Animal-Assisted Interventions: A One Health Approach"

_ijerph, 2019, doi:10.3390/ijerph16193660_

Round 1

Reviewer 1 Report

PLease see attached file

Reviewer 2 Report

I have reviewed many research articles on AAI and have had the opportunity to witness many, many AAI programs in various settings.  The disaointment in the research article is the lack of explanation of the actual intervention (the animals part) and how the animal is paired with the participant. In observing AAI in various settings I have been also disappointed in the lack of standards. Therefore I feel your article is of great importance. The Biophilia concept is spot on.

Line 41 you mentioned pharmaceutical therapy for Alzheimer's, Autism, etc and the importance of non-pharm co-interventions. This is a very vital statement.

Overall the article is excellent, however you mentioned the lack of standardized training. Are you familiar with the Pet Partners program, which started in the 70ies by a psychiatrist and 4 veterinarians?  In this program the animals have to go through a clearance by a vet. The handler takes an 8 hour training on how they should behave such as the animal being bathed before visits, then a test is given. The handler then gets tested with their animal to determine if the animal is suitable.  If the animal passes the animal gets a vest and the handler gets a badge. Various health care facilities or at home clients put in requests for a visit and they are paired based on the clients history and the animal who is determined to be most suited. I do believe most of these are actually visits and not specific to goals and outcomes. In almost all cases the handler is the pets owner. Only one program that I am aware of was done by the late Dr Buettner who if the client had upper body instability and limitations, the client would play fetch with the dog, if they were developing lower body weakness they would take the dog for a walk using a double lead leach, if depressed, a small animal was used to sit in the clients lap and get petted and groomed, and if it was cognition the handler would pose questions and conversation regarding the animal such as what would be a good name for this animal. A second program she developed was having children, who were having difficulty reading, read to a dog.

Just thought you might not be aware of these programs and might consider looking into the pet partners program..petpartners.org

Reviewer 3 Report

Simple summary - I think immediately referencing Hediger et al. is meaningless in this section - it is does not tell the reader anything in a simple form and may put a reader off, if they were looking for some more general insights.I was also surprised that One Health was not mentioned in this, and don't think the summary is very informative.

The paper is very short and rather under developed. I think you could usefully develop the argument in more depth and make it clearer and more convincing. You need to show more clearly the bases on which you are suggesting that animals be chosen for their bonds with humans and could perhaps consider other species in addition to dogs - such as horses for example. And does the kind of AAI matter? How would you assess interspecies bonds? In what cases might this be useful? When/how might it not work? There are some good core ideas here but they are not well developed and consequently the argument is not very convincing. More elaboration and wider use of examples and evidence would improve it on these grounds. 

Reviewer 4 Report

The main aim of this manuscript as stated by authors in their introductory summary is to offer ‘their point of view’ and some suggestions on the methodological approach and the

application of AAIs. Based on my reading of the paper, I concluded that the authors set out to provide a framework defining both the characteristics of the dyadic aspects of relationship functioning between animal and human during AAIs, as well as identifying and operationalizing the influence of a proposed series of variables that would influence this relationship and thus the AAI process or outcome. While I am excited about the aim of the authors – working towards understanding the characteristics and underlying causal mechanisms that underlie effects of AAIs in both species in several domains – as well as provide some guidelines for implementation - there are several issues that limit my enthusiasm about this manuscript. First, given that this is a concept paper, I think it is difficult to put everything into this paper that you may want to address and I would personally suggest that your paper is more a call to action to engage in dialogue recognizing that the points you are making – except for the one about attachment - are already considered by many of the field of HAI – many of these points are talked about in our field. In some way you are ‘preaching to the choir’ and so it would be more powerful to acknowledge that, to say we all know that these issues need to be considered and then call for some action and suggest some specific mechanism to accomplish this. My second issue is about language. I am sorry to say this but throughout the paper, there are several vocabulary and grammar shortcomings and lack of stylistic features common in the English scientific language that detract from easy reading and interpretation. With careful rereading I can figure out what was meant, but the competency with the English language and translated use of terminology detracts from the manuscript. You may be better able to make a compelling call for action if you included a native speaker.

I will discuss what I believe are big picture issue to address.

The overall article is in my opinion mostly a commentary that aims to provide a structure with considerations to take seriously when embarking on research and practice but also implementation and upscaling of animal assisted interventions. The authors essentially make 4 main points and I will comment on each of these and provide some suggestions.

The first is the need to consider the attachment bond between the animal and human in the context of AAI. I am sorry but I disagree; this is not in my opinion the direction our field needs to go. In fact, our field should shy away from what I think are simplistic considerations of complex mechanisms; while attachment may play an important role in some AAI contexts, I would argue that it is but one pathway or dynamic.

The attachment perspective is certainly not new to HAI and in fact, there is some evidence to suggest that it is helpful to consider when studying pet ownership; however, in my opinion, the overall gist in the field is that attachment it is considered less essential or unique when evaluating the quality of interactions during AAIs other than the way Attachment styles inform perceptions and openness about treatment or therapy in general. In this paper, there is no nuanced consideration of the types of AAAs that one may engage in, long term or short term, various species and clients, and so the conclusion that the attachment relationship is the variable that must first and foremost be must be considered is my opinion is not strongly supported by the authors.

In fact, I believe it is incorrect to assume it importance for successful AAIs in general. All one has to consider is for example the work done on college campuses where college students may have the opportunity for short term interaction with dogs right before taking exams or even simply experience the animals’ presence in the room; can we really argue that attachment between in that ‘therapeutic dyad’ must be considered? It may be the case, but I am not convinced that the authors make the case. Similarly, when working with audiences who have PTSD- veterans for example - their attachment style to their primary parent may have played a role in developing PTSD in the context of extreme stress exposure, but this style and the link to how they may interact with an animal during treatment is certainly not the most striking feature of the interaction and it will depend on the setting ownership or therapy. It is just as likely in some cases for some types of interventions that the physiological relaxation – either through petting, stroking or exercising - is what may drive changes in physiology and downregulation of cortisol rather than the ramifications of a working model between the individual and the animals.

Besides the undue importance placed on attachment style conceptually, the authors did not consider recent empirical work done in this realm which actually was based on the belief that attachment would be a central feature of AAI success of efficacy of HAI. For example, work done by Beetz and others that suggested that determining the attachment style systematically and accurately between owners and their dogs was complicated and didn’t provide a predictable framework guiding effective AAIs.

Last, but not least, there are in fact many other mechanisms proposed that have species specific models for treatment efficacy that underlie the unique characteristics of those dyadic interaction bot to mention a few general ones (expectancy, social support etc.) In short, I am not convinced that at this stage in the field, we should identify attachment as the central variable that must be considered.

So, what to do with this section? Recognizing that you have limited space, I suggest reconsidering this main point and making a call for nuance in consideration of exploring potentially mediating or moderating variables. Or, if you feel strongly about this, you will have to make a stronger case for it.

The second point authors are making is that we should focus not one dosage response and merely outcome considerations but also consider the process and its dynamic nature. I fully agree with this point although I do believe that that consideration has already been embraced in the field of HAI minded folks. That said, I don’t think we need to favor one of the other though. Both perspectives are important and represent different research orientations and methods for which there is a room in the field. I also believe that that sentiment is echoes frequently at recent meetings and gatherings of HAI minded folks. I have a similar reaction to your third point. The third issue essentially refers to considering animal wellbeing and again, I think the notion about having to protect animal welfare is one that is frequently communicated by researchers and practitioners alike.

The fourth point is one about zoonotic diseases. Authors state that ‘the zoonotic risk must be seen in the light of the European directive on patient safety and the quality of medical care in order to comply with that the legislation’. While I agree with some of the notions about conducting risk assessments and encouraging safe and thoughtful practice to minimize zoonotic risk, I think this issue is less about ‘inviting’ the states to create collaborations on this issue than it is to encourage sound and thoughtful practice by incorporating some of these standards. The article inadvertently assumes that some of those considerations are not made in the US and while that is not what authors mean – I can recognize the call is for dialogue – the language conveys this less so than you may have intended.

I hope this is helpful as you reconsider your main point and purpose for this paper.

Round 2

Reviewer 1 Report

Please see attached file (copied below too)

Review Paper: ijerph-570658- peer-reveiwed-r1

The human-animal relationship as focus of Animal Assisted Interventions: A One Health approach

Overview:
The authors have done a great job at reconsidering their paper. It reads more fluently and is much more detailed in its arguments.

I have listed only the areas where there is an error or clarification is required. If a section is not listed, it is because it was clear and had no errors- well done!

Abstract:

Ln 32: There is a word missing and the sentence runs onto line 33. “.... give us more data is still lacking on the aspects...” Change to “give us more data. For example, information is still lacking on the aspects...”

Ln: 39: Remove “...and stimuli.” It doesn’t add clarity to the goal of the paper.

AAI Interspecific relationships and attachment bonds

Ln 96: Change “...learns through game, ...” should be games

Ln 99: Change “..., will perform...” to “..., performs...

Suggestions for Research Actions

This section is a big improvement on the previous paper. Now it needs to be tightened up a bit for clarity and for your argument. Some parts were not always clear why you were discussing them. I am not sure why mathematical equations were mentioned and linear arguments. For example, the section’s arguments became clear to me when I got to this part:

“It is precisely that emotional resonance, the variability of the response as well as the unquantifiable and unmeasurable action, which makes AAI an expression of a complex system and it must be considered and studied as such. Within an AAI setting, this interspecific relationship and its circularity influences the results of the intervention, its safety and the welfare of all components (the animal, the operator and the patient). When we consider an AAI setting, the complexity concerns the interrelationship of different variables that, both at an individual and combined level, play a fundamental role in the outcome achievement (e.g. a change  in the patient’s behavior, experiences or characteristics occurring after the intervention) through the action of the dynamic process (e.g. a therapeutic or educational bond, self-relatedness, temporal patterns, etc.). Within an AAI setting we should consider at least four parts of the system - the animal, the operator, the patient and the setting itself.”

It is not essential but some revision of this section remembering the audience could strengthen this paper.

I would suggest the following points for consideration:

AAIs are a complex system with all the parts interacting and influencing each other to make a result that is greater than its parts. The result is improvement in characteristics of the patient. Most medical assessments work with linear systems which lend themselves to assessment in exact, mathematical ways. Therefore, new ways of assessing AAIs are needed.

Consider your arguments simply then write to them. I make this suggestion as this is the one section  of the paper that lost me. Your paper is making important points and needs to take the reader along.

3.1 The AAIs as “complex systems”

Ln 103-104: Clarification: “... systemic approach in the interrelation and reciprocal influence of the relationship...”. There seems to be an important point missing from this sentence. A systemic approach to what aspect of the interrelation and reciprocal influence of the relationship. Is it the evaluation or the study of?

Ln: 103-106: “According to Hediger et al. [21], which reports the need for a systemic approach in the interrelation and reciprocal influence of the relationship between participating humans and animals and included environmental factors and social contexts, we strongly consider and encourage the  study of an interspecific relationship as a reciprocity.” This needs to be split into more than one sentence. Consider “According to Hediger et al. [21], which reports the need for a systemic approach in the interrelation and reciprocal influence of the relationship between participating humans and animals and included environmental factors and social contexts. We strongly consider and encourage the study of an interspecific relationship as a reciprocity.

Ln 129: “ ... , above all ...”  Comma needed “..., above all, ...”

Lns 118-150: In this paragraph the authors refer to AAT and AAI- are they the same thing? If so, use the same term. If not, then a little clarification reminder may be needed.

Ln 146: Clarify: “... effect of a drug is not proportional to the dose; ...”  change to “... effects of some drugs are not proportional to their doses; ...”

3.2 Choice of Animal

Ln 208: Is it just the veterinarian at risk or also the handler and the people taking part in the AAI and AAT?

Ln 223: Spelling error “... his ...” should be “... this ...”

Ln 226: Is it the “... veterinary/dog dyad ...”? Or is it the handler/dog dyad? Most dogs would be handled by non-veterinarians in this situation.

Reviewer 3 Report

The revisions have strengthened the paper. 

Author Response

Thank you for your approval and for your suggestions that have allowed us to improve our manuscript